# Lifestyle interventions delivered by eHealth in chronic kidney disease: A scoping review

Ffion Curtis[1], James O. Burton[2,3], Ayesha Butt[4], Harsimran K. Dhaliwal[5], Matthew M. P. Graham-Brown[2,3], Courtney J. Lightfoot[6], Rishika Rawat[5], Alice C. Smith[6], Thomas J. Wilkinson[4], Daniel S. March[2,3]*

1 Liverpool Reviews and Implementation Group, University of Liverpool, Liverpool, United Kingdom, 2 Department of Cardiovascular Sciences, University of Leicester, Leicester, United Kingdom, 3 John Walls Renal Unit, University Hospitals of Leicester NHS Trust, Leicester, United Kingdom, 4 Leicester Diabetes Centre, University of Leicester, Leicester, United Kingdom, 5 Leicester Medical School, University of Leicester, Leicester, United Kingdom, 6 Department of Population Health Sciences, University of Leicester, Leicester, United Kingdom

* dsm12@leicester.ac.uk

**Data Availability Statement:** All relevant data are within the manuscript and its Supporting Information files.

## Abstract

A method of overcoming barriers associated with implementing lifestyle interventions in CKD may be through the use of eHealth technologies. The aim of this review was to provide an up-to-date overview of the literature on this topic. Four bibliographical databases, two trial registers, and one database for conference proceedings were searched from inception to August 2023. Studies were eligible if they reported a lifestyle intervention using eHealth technologies. A narrative synthesis of the findings from the included studies structured around the type of eHealth intervention was presented. Where a sufficient number of studies overlapped in terms of the type of intervention and outcome measure these were brought together in a direction of effect plot. There were 54 included articles, of which 23 were randomised controlled trials (RCTs). The main component of the intervention for the included studies was mobile applications ($n = 23$), with the majority being in the dialysis population ($n = 22$). The majority of eHealth interventions were reported to be feasible and acceptable to participants. However, there was limited evidence that they were efficacious in improving clinical outcomes with the exception of blood pressure, intradialytic weight gain, potassium, and sodium. Although eHealth interventions appear acceptable and feasible to participants, there is insufficient evidence to make recommendations for specific interventions to be implemented into clinical care. Properly powered RCTs which not only demonstrate efficacy, but also address barriers to implementation are needed to enhance widespread adoption.

## Introduction

For individuals living with chronic kidney disease (CKD), having a healthy lifestyle (e.g. being physically active, consuming a healthy diet, and not smoking) can slow disease progression and reduce both cardiovascular risk and all-cause mortality [1–3]. For these reasons adopting a healthy lifestyle is recommended by clinical practice guidelines for this population [4–6].

**Funding:** The author(s) received no specific funding for this work.

**Competing interests:** The authors have declared that no competing interests exist

These recommendations are supported by recent randomised controlled trial (RCT) data showing that a 36-month lifestyle intervention doubled the number of individuals with CKD who were able to meet physical activity guidelines [7]. Despite this evidence, there are no interventions to promote a healthy lifestyle embedded as part of normal clinical care for individuals living with CKD. Some of the challenges to implementing lifestyle interventions in clinical practice include their resource-intensive nature (and subsequent funding limitations) [7], issues around effectiveness, accessibility, feasibility with translating research findings into real world settings [8], and the lack of cost-effective analyses [9]. Furthermore, CKD is more common in older age, and is associated with multiple long-term conditions [10], and low levels of health literacy [11], which further complicates the implementation process. Further research is still needed with particular focus on how to translate lifestyle interventions of proven efficacy into clinical practice in a sustainable way.

One innovative method of delivering healthy lifestyle interventions that may overcome the identified barriers to implementation [8] is through technology-based electronic health (eHealth) interventions defined as "health services and information delivered or enhanced through the internet and related technologies" [12]. Lifestyle interventions containing a number of components may be more suited to delivery through eHealth particularly for those individuals with early stages CKD where promoting behaviour change may have a positive impact on disease progression and outcome [3]. A Cochrane review from 2019 [13] reported that eHealth interventions aiming to promote behaviour change in CKD may improve the management of dietary sodium intake and fluid management. The overall findings were limited due to the heterogeneity and low quality of available evidence. However, participants did report high levels of satisfaction due to the eHealth interventions being informative, of low burden, and easy to use [13]. There is a growing interest in the use eHealth and digital technologies for the delivery of healthcare in CKD, particularly following the rise of digital healthcare utilisation during COVID-19, and thus an up-to-date scope of the evidence in this population is required [14, 15].

The research aims of this review were to:

- Identify the mode of eHealth interventions that have been employed to deliver lifestyle interventions in the CKD population within the existing literature.

- Provide a summary overview of the effect of these interventions on a range of outcomes as reported in the primary studies.

## Materials and methods

A scoping review was chosen as there has been a growth in eHealth interventions since the COVID-19 pandemic, therefore we wished to provide an overview from the existing literature on the mode and effect of delivery of these technologies in the lifestyle context within the CKD population. We followed the preferred reporting items for systematic reviews and meta-analyses extension for scoping reviews (PRISMA-ScR) checklist [16] (S1 Checklist) and the Arksey & O'Malley [17] framework for conducting this review. This review was prospectively registered on Figshare: https://figshare.com/articles/preprint/EHealth_and_Lifestyle_Review_Protocol_v1_1_16122022/21842958. With the last protocol update on the 16th of December 2022.

### Inclusion criteria

Studies were eligible if they reported a lifestyle intervention using eHealth technologies. eHealth technologies include: mobile applications, computer and tablet based applications,

personal digital assistants, web-based/internet applications, virtual reality tools and other eHealth technologies as defined by the CONSORT eHealth Group [18].

More specifically, the inclusion criteria were:

- Participants: Individuals (including paediatric) with all stages of CKD (including those with end-stage kidney disease (ESKD) receiving kidney replacement therapy). We included those individuals with an episode of acute kidney injury (AKI).

- Concept: The review considered studies that include any combined lifestyle intervention (e.g. physical activity or exercise, dietary (including modifying sodium and protein intake), weight loss and reducing alcohol intake) or lifestyle component delivered alone (e.g. physical activity only) via eHealth in the CKD.

- Setting: Lifestyle eHealth interventions could be delivered in a number of settings including primary care, clinics, haemodialysis units or rehabilitation services.

- Types of study: This scoping review considered the following study design (but not limited to): interventional studies (randomised and non-randomised controlled trials, quasi-randomised studies), observational studies (e.g. cohort and cross-sectional studies), qualitative studies, and process evaluations relating to eHealth interventions. Published protocols of studies and conference abstracts were included. Studies that included individuals with CKD and caregivers or family members were excluded.

## Search strategy

The following bibliographical databases and trial registers were searched for completed and ongoing studies: MEDLINE, EMBASE, CINAHL, Cochrane Central Register of Controlled Trials (CENTRAL); trials only, ClinicalTrials.gov, and the World Health Organisation International Clinical Trials Registry Platform. Conference Proceedings Citation Index (Web of Science™ Core Collection) were searched for unpublished data. All databases were searched from inception to 2nd August 2023, with no limits on language set. Database searches were supplemented with a Google Scholar search with the first 200 titles screened for inclusion [19]. An example of a full search strategy for MEDLINE is presented in S1 File.

## Study selection and data charting

Search results were compiled using the web-based screening and data extraction tool Covidence (Veritas Health Innovation Ltd., Melbourne, Australia). Duplicate records were removed and a two-part study selection process was used: (first part) title and abstracts were screened independently by two reviewers against the inclusion criteria, and (second part) full-text articles not excluded based on title or abstracts were retrieved and assessed by two reviewers. If there was a disagreement then this was resolved through the inclusion of a third reviewer.

## Data extraction and synthesis

We developed, tested, and refined a structured data collection form based on the Cochrane Data Extraction Template for interventions but with modifications for non-interventional studies. One reviewer undertook data extraction for each study, with a second reviewer (DSM) cross checking all extracted data. For each included study, characteristics including study design, CKD population, sample size, lifestyle component, type of eHealth intervention, and

main outcomes were extracted and presented in table format. As were the eHealth intervention description, type of comparison, length of follow-up and main findings.

We created a narrative synthesis of the findings from included studies structured around the type of eHealth intervention reported (e.g. mobile applications, short message service (SMS), videoconferencing/online videos, virtual reality exercise (VREx), and web-based platforms)). Data were presented as text, tables, and where a sufficient number of studies overlapped in terms of type of eHealth intervention, and outcome measure, then we brought these together visually in a direction of effect plot [20, 21]. For the direction of effect plots the direction and magnitude of (clinically meaningful) change along with the statistical significance from within each study were considered to indicate either a positive health effect, negative health effect, or no effect [20].

## Quality assessment

Whilst quality assessment of the included studies is not an essential component of a scoping review, we chose to include it because we felt that it would add value, potentially providing the basis for recommendations for future research with regard to the design, conduct and reporting. We quality assessed full-text articles of studies ($n$ = 37) with the exception of conference abstracts, developmental studies and process evaluations. The National Institutes of Health (NIH) Quality Assessment Tool for Controlled Intervention Studies was used to assess the quality of each included study (available from https://www.nhlbi.nih.gov/health-topics/study-quality-assessment-tools). Please see S1 Table for the criteria used to assess the quality of the included studies. Two reviewers evaluated the quality of the included studies independently. Overall quality rating was rated as poor (0–4 as "yes"), fair (5–10 as "yes"), or good (11–14 as "yes") for the controlled intervention studies (RCTs), and the non-RCTs) [22]. If there was a "fatal flaw" defined as high dropout rates (question 7), high differential dropout rates (question 8), no intention to treat analysis (question 14) then it was downgraded a category (regardless of overall score). The overall rating for the cohort and cross-sectional was scored in the same way as the controlled intervention studies (with no "fatal flaw" questions). The uncontrolled before-after studies were rated as poor (0–4 as "yes"), fair (5–8 as "yes"), or good (9–12 as "yes"), a "fatal flaw" was defined as not sufficiently large sample size (question 5), or >20% loss to follow-up (question 9).

## Results

Fig 1 provides a flow diagram of article identification and inclusion. There was one study in Korean [23], which contributed only limited information [24] (S2 Table). Seven trial registrations were excluded from the narrative synthesis as they did not contain enough information (S2 Table). Two articles [25, 26] reported the same study but in different journals, and a further trial was published as two reports in the same journal [27, 28]. This left 54 individual articles (52 separate studies). (Fig 1).

## Characteristics of included reports

Of the 54 articles, there were 23 RCTs (one reported as a conference abstract). Intervention duration ranged from 4 to 52 weeks, and included 1,682 randomised participants. There was 15 non-randomised studies (one study was reported twice [25, 26]) (this included eight uncontrolled before-after studies, four non-RCTs, two cohort studies and one cross-sectional study), four process evaluations (plus a further evaluation [28] of an included RCT [27]) and two developmental studies [29, 30] (Table 1). Lastly, there were eight protocol publications of RCTs in progress [31–38] (S3 Table). The eHealth component for 22 of these primarily

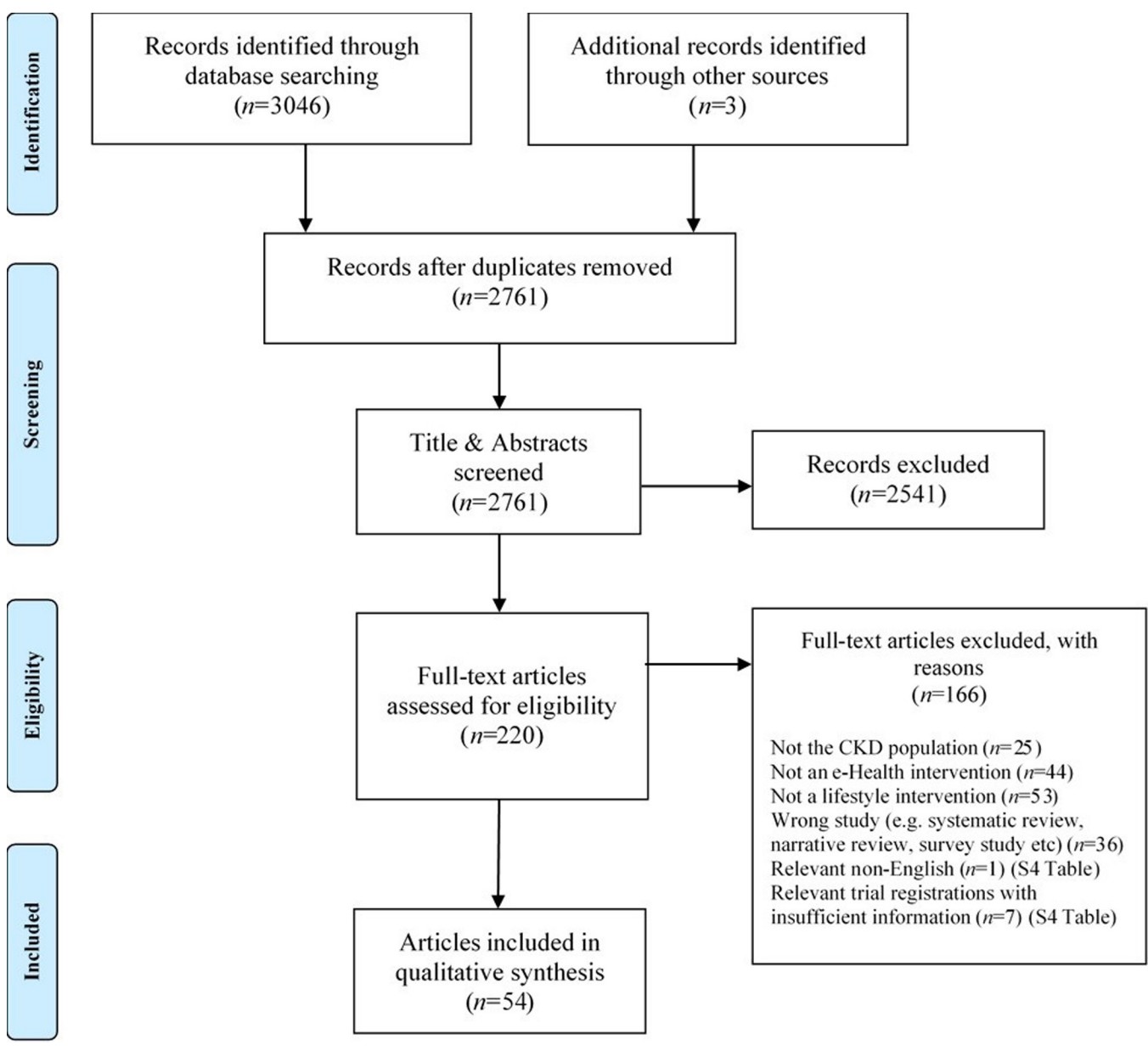

**Fig 1. Flow diagram of article selection.**

involved mobile applications, seven involved web-based platforms, six involved SMS messages, six included videoconferencing or online videos, five included VREx, three included wearables, one was web-based messaging, and the remaining two involved a personal digital assistant (PDA), and tablet-based application (Tables 2 and S3). There were 22 studies/evaluations in the dialysis population, 18 in the CKD stages 1–5 population and a further nine in transplant recipients (including one in the paediatric population). In addition, there were two in the CKD and ESKD population, and one in individuals with either CKD or ESKD (dialysis and transplant) (Tables 1 and S3). There were no studies in the AKI population. Twenty-two studies/evaluations involved lifestyle (including dietary and physical activity) interventions,

**Table 1. Characteristics of included studies, process evaluations and developmental studies.**

| Study | Country | Study design | CKD population and Sample Size | Main lifestyle component | Type of E-health interventions | Main outcomes |
|---|---|---|---|---|---|---|
| Mobile Applications | | | | | | |
| Bruen et al., 2022 [39] | USA | Process evaluation | CKD Stages 2–4; n = 24 | Lifestyle | Mobile application + videoconferencing (Zoom) + social media. | Carbohydrate intake, blood glucose, blood insulin, blood ketone, animal protein intake, diet quality, adherence feasibility and satisfaction. |
| Chiang et al., 2021 [40] | Taiwan | NRCT | Haemodialysis; n = 65; Intervention n = 30; Control n = 30 | Dietary | Mobile Application for 12 weeks. | Self-efficacy, serum phosphate, serum albumin. |
| Doyle et al., 2019 [41] | Ireland | UCBA | CKD stages 2–5; n = 23 | Lifestyle | Mobile Application (MiKidney) for 12 weeks. | Physical Activity (International Physical Activity Questionnaire), 6-minute walk test, dietary intake and clinical outcomes. |
| El-Khoury et al., 2020 [26] & 2021 [25]* | United Arab Emirates | UCBA | Haemodialysis; n = 26 | Dietary | Mobile Application (Kidney Education for Lifestyle Application) for 2 weeks. | Acceptability and feasibility, adherence to dietary guidelines, BMI, body weight, dietary intake. |
| Hayashi et al., 2017 [42] | Japan | NRCT | Haemodialysis; n = 20; Intervention n = 9; Control n = 11 | Dietary | Mobile Application (SMART-D) for 2 weeks. | Feasibility and usability, intradialytic weight gain, serum potassium, serum phosphorus, and HRQoL. |
| Li et al., 2020 [43] | Taiwan | RCT | CKD Stages 1–4; n = 60; Intervention n = 30; Control n = 30 | Lifestyle | Mobile Application + Wearable for 90 days. | Self-efficacy, self-management, HRQoL, steps per day and clinical outcomes. |
| Lin et al., 2014 [44] | Taiwan | CSS | CKD Stage 1–5; n = 20 | Dietary | Mobile Application. | Acceptability and Usability. |
| Liu et al., 2023 [45] | China | CS | CKD Stages 1–4; n = 2060; Intervention n = 1600; Control n = 460 (Propensity score matching) | Lifestyle | Mobile application + web-based clinical dashboard for health coaching for a mean follow up of 18.1 months. | 30% decrease in eGFR or incidence of ESKD, 24h proteinuria and mean arterial pressure (MAP). |
| Kowal et al., 2023 [46] | Poland | Process evaluation | Haemodialysis; n = 60 | Dietary | Mobile application | N/A |
| Pack et al., 2021 [47] | South Korea | RCT | Haemodialysis; n = 84; Intervention n = 42; Control n = 42 | Dietary | Mobile Application + face-to-face training x 3 per week for 8 weeks. | Serum phosphorus, serum potassium, serum albumin, self-efficacy, and HRQoL. |
| Pinto et al., 2020 [48] | Brazil | UCBA | Haemodialysis; n = 52 | Dietary | Mobile Application (NefroPortail App) for 12 weeks. | HRQoL, self-management of disease, serum calcium and clinical outcomes. |
| Pollock et al., 2023 [49] | USA | RCT | Transplant recipients; n = 16; Intervention n = 7; Control n = 9 | Lifestyle | Mobile application (MyKidneyCoach) + Nurse Coaching. | Feasibility and acceptability, patient activation measure (PAM), nutrition self-efficacy, immunosuppression and clinical outcomes |
| Schrauben et al., 2022 [50] | USA | UCBA | CKD Stages 1-3a; n = 44 | Dietary | Mobile Application (www.myfitnesspal.com )+ Tele-counselling, daily e-mails, and access to a study website for 8 weeks. | 24-hour urine sodium; healthy eating index; protein intake, 24-hour urine albumin, urine potassium, adherence, body mass, SBP and DBP. |

(*Continued*)

**Table 1.** (Continued)

| Study | Country | Study design | CKD population and Sample Size | Main lifestyle component | Type of E-health interventions | Main outcomes |
|-------|---------|--------------|-------------------------------|--------------------------|--------------------------------|---------------|
| St-Jules et al., 2023 [51] | USA | RCT | CKD Stages 1–4; *n* = 256; Intervention 1 *n* = 64; Intervention 2 *n* = 64; Intervention 3 *n* = 64 | Lifestyle | Intervention 1 = Weekly videoconferencing sessions for 24 weeks; Intervention 2 = Weekly videoconferencing sessions + Mobile Application for 24 weeks; Intervention 3 = Weekly videoconferencing sessions + Mobile Application for 24 weeks. | Body mass, body mass loss, urine sodium, urine phosphorus and clinical outcomes. |
| St-Jules et al., 2021 [52] | USA | RCT | Haemodialysis; *n* = 40; Intervention 1 *n* = 14; Intervention 2 *n* = 13; Control *n* = 13 | Dietary | Intervention 1 = mobile application for 24 weeks; Intervention 2 = mobile application + online videos for 24 weeks. | Feasibility, acceptability, serum phosphorus, and serum albumin. |
| Teong et al., 2022 [53] | Malaysia | RCT | Haemodialysis; *n* = 74; Intervention *n* = 38; Control *n* = 36 | Lifestyle | Mobile application for 12 weeks. | Serum phosphorus, phosphorus intake, knowledge of phosphorus management, phosphate binder adherence, and clinical outcomes. |
| Torabikhah et al., 2023 [54] | Iran | RCT | Haemodialysis; *n* = 70; Intervention *n* = 35; Control *n* = 35 | Lifestyle | Mobile application for 4 weeks. | Intradialytic weight gain, potassium, phosphorus, cholesterol, triglyceride, aluminium and ferritin. |
| Tsai et al., 2021 [55] | Taiwan | CS | CKD Stages 1–5; *n* = 268; Intervention *n* = 134; Control *n* = 134 (Propensity score matching) | Lifestyle | Mobile application for 3 months. | Disease Knowledge and Self-care behaviours. |
| Welch et al., 2013 [56] | USA | RCT | Haemodialysis; *n* = 44; Intervention n = 24; Control *n* = 20 | Dietary | Mobile application for 6 weeks. | Usual care + daily activity monitoring application on a personal digital assistant |
| Short-messaging service (SMS) | | | | | | |
| Arad et al., 2021 [57] | Iran | RCT | Haemodialysis; *n* = 66; Intervention n = 33; Control *n* = 33 | Lifestyle | SMS text messages x1 per day for 12 weeks. | Haemodialysis adherence, medication adherence, fluid restrictions, diet recommendations, and clinical outcomes. |
| Bruinius et al., 2022 [58] | USA | RCT | CKD Stages 3; *n* = 36; Intervention *n* = 19; Control *n* = 17 | Lifestyle | SMS text messages (x1 per day) + one on one dietitian-led educational session. | Acceptability and feasibility. |
| Cueto-Manzano et al., 2015 [59] | Mexico | UCBA | (>14 years) Transplant recipients; *n* = 23 | Lifestyle | SMS text messages x 50. | Usefulness. |
| Dawson et al., 2021 [60] | Australia | RCT | Haemodialysis; *n* = 130; Intervention n = 87; Control *n* = 43. 2:1 Allocation | Lifestyle | SMS text messages (x3 per week.) | Acceptability, feasibility, adherence to dietary recommendations and clinical outcomes. |
| Kelly et al., 2019 [28] & 2020 [27]* | Australia | RCT | CKD stages 3–4; *n* = 80; Intervention *n* = 41; Control *n* = 39 | Dietary | SMS text messages every 2 weeks for 6 months + telephone calls every 2 weeks for 3 months. | Feasibility, acceptability, diet quality, and clinical outcomes. |
| Modanloo et al., 2015 [61] | Iran | RCT | Haemodialysis; *n* = 70; Intervention *n* = 35; Control *n* = 35 | Lifestyle | SMS text messages (x 6 per week for 6 week) + 2 in person training sessions. | Body mass and participant satisfaction. |
| Videoconferencing/online videos | | | | | | |
| Begue et al., 2022 [78] | USA | RCT (Conference Abstract) | CKD Stages 3b-5; *n* = 17; Intervention *n* = 12; Control *n* = 5 | Physical activity | Videoconferencing supervised exercise for 12 weeks. | Cardiorespiratory fitness, 6-minute walk test. |

*(Continued)*

**Table 1.** (Continued)

| Study | Country | Study design | CKD population and Sample Size | Main lifestyle component | Type of E-health interventions | Main outcomes |
|---|---|---|---|---|---|---|
| Gibson et al., 2020 [62] | USA | RCT | Transplant recipients; *n* = 10; Intervention *n* = 5; Control *n* = 5 | Lifestyle | Videoconferencing for 12 weeks. | Feasibility and acceptability, medication use, anthropometrics, blood pressure, physical activity, HRQoL, dietary intake. |
| Leal et al., 2022 [63] | Portugal | Process evaluation (Conference Abstract) | Haemodialysis; *n* = 2063 | Physical Activity | Videoconferencing. Online exercise programme via Zoom®. | Programme adoption, adherence and physical function. |
| Ravaglia et al., 2019 [64] | Italy | Process Evaluation (Conference Abstract) | Haemodialysis, Peritoneal Dialysis; Transplant Recipients; *n* = 1408 | Physical Activity | Online exercise videos + mobile applications | Physical function, death, cardiovascular events, falls, fractures and hospitalisations. |
| Virtual reality exercise | | | | | | |
| Chou et al., 2020 [65] | Taiwan | NRCT | Haemodialysis; *n* = 64; Intervention *n* = 32; Control *n* = 32 | Physical activity | Virtual reality exercise x 3 per week for 30 minutes for 4 weeks. | Fatigue. |
| Maynard et al., 2019 [66] | Brazil | RCT | Haemodialysis; *n* = 45; Intervention *n* = 22; Control *n* = 23 | Physical activity | Virtual reality exercise + intradialytic cycling x 3 per week for 30–60 minutes for 12 weeks. | Gait speed, timed up and go, physical activity, HRQoL and depression. |
| Segura-Ortí et al., 2019 [67] | Spain | RCT | Haemodialysis; *n* = 36; Intervention *n* = 18; Control *n* = 18 | Physical activity | Virtual reality exercise x 3 per week for 30 minutes for 4 weeks | 6-minute walk test, gait speed, sit-to-stand 10, sit-to-stand 60, one-leg heel-rise, adherence. |
| Weigmann-Faßbender et al., 2020 [68] | Germany | NRCT | Paediatric Transplant Recipients; *n* = 21; Intervention *n* = 13; Control *n* = 8 | Physical Activity | Virtual reality exercise programme; 3x30 minutes per week for 6 weeks. | Physical activity; $VO^2$ peak, hand grip strength and HRQoL. |
| Zhou et al., 2020 [69] | Qatar | RCT | Haemodialysis; *n* = 73; Intervention *n* = 37; Control *n* = 36 | Physical activity | Virtual reality exercise X 3 per week for 30 minutes for 4 weeks. | Depression & user experience. |
| Web-based platforms | | | | | | |
| Castle et al., 2022 [70] | United Kingdom | RCT | Transplant recipients; *n* = 17; Intervention *n* = 9; Control *n* = 8 | Lifestyle | Web-based Platform for 12 weeks. | Feasibility outcomes (including screening, retention, engagement and experience), and clinical outcomes. |
| Donald et al., 2022 [71] | Canada | UCBA | CKD Stages 3a-5; *n* = 33 | Lifestyle | Web-based platform (My Kidneys My Health) for 8 weeks. | Acceptability, usability, and self-efficacy. |
| Heiden et al., 2013 [29] | Denmark | Development and Qualitative study | CKD, haemodialysis and transplant recipients; *n* = 5 | Dietary | Web-based platform | Usability and testing. |
| Humalda et al., 2020 [72] | Netherlands | RCT | CKD Stages 1–4; *n* = 99; Intervention *n* = 52; Control *n* = 47 | Dietary | Web-based platform for self-management+ e-coaching (by telephone or e-mail) for 36 weeks. | Sodium excretion, blood and urinary electrolytes, blood pressure, proteinuria, HRQoL, self-management skills, barriers and facilitators to implementation. |
| Ong et al., 2022 [30] | Canada | Development and Qualitative study | CKD stages 3b-5; *n* = 11 | Lifestyle | Web-based platform | Satisfaction. |
| Other eHealth Interventions | | | | | | |
| Anand et al., 2021 [73] | USA | RCT | CKD stages 3b-4; *n* = 64; Intervention *n* = 32; Control *n* = 32 | Physical Activity | Wearable (Garmin Vivofit 3) for 8 weeks + Mobile Application. | Physical activity, feasibility, physical function, blood pressure, anthropometrics, self-reported mental health, self-efficacy. |

(*Continued*)

**Table 1.** (Continued)

| Study | Country | Study design | CKD population and Sample Size | Main lifestyle component | Type of E-health interventions | Main outcomes |
|---|---|---|---|---|---|---|
| Naseri-Salahshour et al., 2020 [74] | Iran | RCT | Haemodialysis; *n* = 104; Intervention *n* = 52; control *n* = 52. | Dietary | Web-based messages (e-coaching) x 2 per week for 4 weeks. | HRQoL, serum sodium, serum phosphorus, serum potassium, serum calcium, serum magnesium. |
| O'Brien & Meyer., 2020 [75] | USA | UCBA | Transplant recipients; *n* = 53 | Physical activity | Wearable activity tracker (Fitbit) for 4 weeks. | Feasibility and acceptability. |
| Sevick et al., 2016 [76] | USA | RCT | Haemodialysis; *n* = 191 (*n* = 12 dropped out before baseline assessment); Intervention *n* = 93; control *n* = 86 | Dietary | Personal digital assistant x 3 meals a day and 1 snack for 16 weeks. | Interdialytic body mass gain and dietary sodium. |
| Zemp et al., 2022 [77] | Switzerland | UCBA | Haemodialysis; *n* = 22 | Physical Activity | Tablet-based application to deliver and record the exercise programme + initial face to face training sessions for 12 weeks. | 44% (n = 86) of participants were eligible. Out of these n = 22 agreed to participate. Adherence was 73%. No significant change in physical function. |

Chronic kidney disease (CKD), cohort study (CS), cross sectional study (CST), health related quality of life (HRQoL), non-randomised controlled trial (NRCT), not applicable (N/A), randomised controlled trial (RCT), short message service (SMS), uncontrolled before-after study (UCBA). *Reports of the same study so have been grouped for synthesis.

with a further 15 including physical activity and dietary interventions respectively (Tables 1, 2 and S3).

## Quality rating for the included RCTs

Quality rating summaries are provided in Figs 2–4. Four of the included RCTs & non-RCTs were rated as "good" quality with the remaining rated as either "fair" or "poor" (Fig 2). All the uncontrolled before-after studies, cohort and cross-sectional studies were rated as either "fair" or "poor" (Figs 3 and 4).

## Mobile applications

There were eight RCTs [43, 47, 49, 51–54, 56], nine non-randomised study designs [25, 26, 40–42, 44, 45, 48, 50, 55] and two process evaluations [39, 46] of dietary or lifestyle interventions primarily delivered by mobile applications on a range of outcomes (Table 1). There were six reports of feasibility/usability [26, 42, 44, 48, 49, 52], five for self-efficacy [40, 43, 47, 49, 56] and acceptability [42, 44, 49, 52, 56], and four for health related quality of life (HRQoL) [42, 43, 47, 48], with positive effects generally reported for these outcomes (Table 3). Similarly, there were three reports on knowledge [25, 40, 53], two for self-management [43, 48] and one for disease knowledge [55] with positive effects reported on these outcomes (Table 3). Seven reported serum phosphorus [25, 42, 47, 48, 52–54], one serum phosphate [40], two urine phosphorus [50, 51], and another dietary phosphorus intake [56] (Table 1). With most reporting no change (Table 4). Five reported serum albumin [26, 40, 47, 52, 53], with another reporting urine albumin [50], there was no effect on this outcome (Table 4). There were five reports of serum potassium [26, 42, 47, 48, 54], one for dietary potassium intake [56], and one for urine potassium [50] with the majority reporting a positive effect (Table 4). For sodium there were

**Table 2. Characteristics of study interventions and outcomes.**

| Study | eHealth intervention description | Type of Comparison | Length of Follow-up | Main findings |
|---|---|---|---|---|
| Mobile Applications | | | | |
| Bruen et al., 2022 | Ren Nu is a dietitian-led and supervised, remote training program in a class setting of approximately 20 participants per class. The program is facilitated by digital technology (Zoom (Zoom Video Communications, San Jose, USA) and Facebook (Facebook, Menlo Park, California, USA) to enable teaching, communication, as well as tracking and monitoring of dietary data and blood pressure. | N/A | 12 weeks | Participants reported reductions in BMI, there was also reductions in blood pressure and biomarkers. Overall satisfaction with diet was rated as high. |
| Chiang et al., 2021 | The mobile phosphate control application allowed users to search for items or food additives by their formal or common names in Mandarin or by pictures. The application also provided users with useful recipes and cooking tips. | N/A | 12 weeks | Significant increase in self-efficacy and knowledge. No difference in blood biochemistry |
| Doyle et al., 2019 | The MiKidney app records personal details, medical history, bloods, weight and current medication list. It provides information on medication, diet, symptom management and staying healthy. The app includes an exercise tracker. | N/A | 12 weeks | Significant increases in six-minute walk test. Significant decreases in waist circumferences and circulating cholesterol. |
| El-Khoury et al., 2020 & 2021* | Designed to provide dietary education and traditional renal diet–friendly recipes to haemodialysis patients. | N/A | Post 2 weeks app usage | Decrease in self-reported non-adherence, improvement in knowledge, increase in dietary protein intake. |
| Hayashi et al., 2017 | e-Self-management and recording System for Dialysis (SMART-D), a simple smartphone-based system focused on intradialytic weight gain, potassium, and phosphorus. | N/A | 2 weeks | All 9 patients in the SMART-D group were able to complete the 2-week use of the system without any major problems. Significant increase in HRQoL. |
| Li et al., 2020 | Wearable wristband detecting steps, calories and heart rate to a mobile application (WowGoHealth). A mobile application (LINE, LINE Corporation) to record dietary data. | Usual care | 90 days | Significant increase in self-efficacy and self-management. No difference in body weight, cholesterol or triglyceride. |
| Lin et al., 2014 | The proposed app includes information on going healthcare issues and dietary considerations. In addition to maintenance of stable blood sugar values, blood pressure control and weight management. | N/A | N/A | All participants found the app easy to use, acceptable reliable and functional. |
| Liu et al., 2023 | The system included interpretation of disease and guidance on diet and exercise, check-ups, warnings, reminders and questions and answers from health coaches | Usual care + access to the health-related educational materials in the app only | 18.1 ± 9.5 months | Primary composite kidney outcome occurred in n = 121 (8%) participants in the intervention group and n = 33 (7%) in the conventional care group. MAP and proteinuria improved in both groups. |
| Kowal et al., 2023 | The app provides a database of the nutritional content of dietary products and renal recipes | N/A | N/A | There were high satisfaction ratings reported for the app. |
| Pack et al., 2021 | Self-management of diet through application. Participants entered type and amount of food they ate the application screen displayed the simulated values of calories, protein, phosphorus, sodium, potassium and albumin. | Usual care + face-to-face training | 12 weeks | Significant decrease in serum phosphorus and potassium. Significant increase in self-efficacy and HRQoL. |
| Pinto et al., 2020 | A mobile app that helps participants control their intake of liquids and foods. Food consumption was entered daily and reviewed nutritionist. | N/A | 12 weeks | No significant difference in interdialytic weight gain, Decrease in phosphorus levels and self-management. Usability was rated as very positive. |

(*Continued*)

**Table 2.** (Continued)

| Study | eHealth intervention description | Type of Comparison | Length of Follow-up | Main findings |
|-------|----------------------------------|--------------------|---------------------|---------------|
| Pollock et al., 2023 | The mobile app included educational videos and newsletters including symptoms, complications, appointments, medication, nutrition, hydration, exercise and emotional health. Along with monitoring tools for medication, blood pressure, exercise, diet and sleep. | Usual care + attention control | 12 weeks | Did not meet the target sample size. 50% participation rate, acceptability 67.5%, participants that remained in the study had higher PAM scores. PAM and self-efficacy increased in both groups. Improvement in hypertension in the intervention group. |
| Schrauben et al., 2022 | Participants used the app to self-monitor their diet daily. Dietitians then reviewed their dietary intake and provided feedback. Participants also accessed a study-developed educational website. | N/A | 12 months | The 24-hour mean urine sodium did not decline over the study period. The mean healthy eating index improved. No change in albumin. |
| St-Jules et al., 2023 | Intervention 1 = group-based videoconferencing which were dietary educational and social cognitive theory based behavioural counselling; Intervention 2 = Group based educational only videoconferencing, and a food logging mobile application directed to record all food and beverage intake, as well as their physical activity and body weight; Intervention 3 = all components of Intervention 1 & Intervention 2. | Usual care + written information about dietary intervention targets. Monthly 1-page educational handouts | 24 weeks | No change in body mass. No change in dietary sodium. No change in phosphorus intake. |
| St-Jules et al., 2021 | Intervention 1 = Food logging mobile application (MyNetDiary, Inc., Marlton, NJ, USA) directed to record all food and beverage intake and monitor phosphorus content of food 2 = Food logging mobile application (MyNetDiary, Inc., Marlton, NJ, USA) and social cognitive theory-based counselling through online videos provided by Powtoon (Powtoon Ltd., London, UK). | Educational videos only (iPad) limiting phosphorus intake, maintaining protein intake and adherence to phosphorus intake | 24 weeks | Participants reported the intervention to be feasible and acceptable. No significant differences in serum albumin or phosphorus. |
| Teong et al., 2022 | Phosphate mobile app (PMA), MyKidneyDiet-mobile application Phosphate Tracker ©2019 | Usual care + single face to face dietitian led session on hyperphosphatemia | 12 weeks | No change in serum phosphorus. There was a significant between group change in serum calcium. No change in serum albumin. |
| Torabikhah et al., 2023 | The mobile application contained educational videos. Topics included diet, fluid, physical activity and medication. The app could also receive messages. | Usual care + face to face training | 16 weeks | Significant increases in the intervention group for intradialytic weight gain, potassium, phosphorus, cholesterol, and triglyceride. |
| Tsai et al., 2021 | A healthcare mobile application, called iCKD, which has ten major features, participants enter information pertaining to blood pressure, blood sugar, exercise, dietary data and medication compliance. | N/A | 12 weeks | The patients who used iCKD had higher disease knowledge scores than those who did not use iCKD. |
| Welch et al., 2013 | A mobile app containing a nutrition database allowing individuals to scan food packages and receive feedback on intake in relation to their dietary prescriptions. | Usual care + daily activity monitoring application on a Personal Digital Assistant | 6 weeks | No change in intradialytic weight gain. No difference in perceived benefits or self-efficacy. The intervention was acceptable. |
| Short-messaging service (SMS) | | | | |
| Arad et al., 2021 | The text message concerned patient education topics on diet, medication use and fluid restriction. | Usual care | 12 weeks | Significant increase in adherence There was also significant changes in medication adherence, fluid restriction and diet recommendation. |

*(Continued)*

**Table 2.** (Continued)

| Study | eHealth intervention description | Type of Comparison | Length of Follow-up | Main findings |
|---|---|---|---|---|
| Bruinius et al., 2022 | Daily dietary SMS messages. The messages concerned patient education topics on diet (low sodium, reading food labels), medication use, and fluid restrictions.) | Usual care + one on one dietitian led educational session | 4 weeks | Participants in the intervention group reported viewing >90% of received text messages. Participants found the texts to be useful. |
| Cueto-Manzano et al., 2015 | 50 SMS text messages about kidney disease risk factors, 40 about healthy lifestyle and 35 recommendations to improve adherence to treatment, attendance at follow-up appointments. | N/A | Median duration of 58 days | At the end of the study, participants found the text messages useful. They perceived the tool as helpful and considered that it could be implemented widely. |
| Dawson et al., 2021 | SMS messages contained advice, information, and motivation and support to improve renal dietary behaviours and general healthy eating and lifestyle behaviours. | Usual care | 24 weeks | The intervention was acceptable, feasible and participants were adherent. |
| Kelly et al., 2019 & 2020* | SMS messages included educational, self-monitoring and goal setting. | Usual care + SMS text messages for 3 months | 24 weeks | Recruitment rate was 35%. 95% completed the programme. Overall participant experience of the intervention was positive. Significant improvements in diet quality, body mass and cost effective. |
| Modanloo et al., 2015 | SMS messages were associated with self-care behaviours for weight control, fluid and salt restriction, medicine, exercise and nutrition tips. | Usual care + 2 in-person training sessions | 6 weeks | Significant decrease in body mass. 80% of participants reported that the intervention was acceptable. |
| Videoconferencing/online videos | | | | |
| Begue et al., 2022 | Each one-week video-conferencing session was conducted by exercise trainers using a videoconference tool. | Diet and Exercise counselling at baseline | 12 weeks | Increase in total work on the cardiorespiratory fitness test and increase in 6-minute walk test score. |
| Gibson et al., 2020 | Participants had 1 hour health coaching sessions including portion size education, cooking skills and were encouraged to accumulate 150 minutes of activity a week. | Usual care + healthy lifestyle tracking | 24 weeks | Adherence rate of the health coaching was 78%. Retention rate was 100%. Significant changes in body mass No change in BP, physical activity or HRQoL. |
| Leal et al., 2022 | Online exercise programme | N/A | 12 weeks | The programme was adopted by 16 units (66.7%) and 4%. Adherence was 73.1 ±18.8%, and participants improved performance in all physical function measures. |
| Ravaglia et al., 2019 | The exercise program will be explained by digital technology including videos and app for smartphone | N/A | Process evaluation (ongoing) | Physical functional, the incidence of death, cardiovascular events, falls, fractures and hospitalisation days assessed every 3 and 6 months. |
| Virtual Reality Exercise | | | | |
| Chou et al., 2020 | Virtual reality exercise programme using Nintendo® Wii outside of dialysis. | Usual care | 4 weeks | Overall fatigue declined in both groups. No difference between groups in clinical characteristics. |
| Maynard et al., 2019 | Virtual reality exercise programme during dialysis using a number of Nintendo® Wii games. | Usual care | 12 weeks | Significant increase in timed up and go, the Duke Activity Status Index and HRQoL. |
| Segura-Orti et al., 2019 | Virtual reality exercise programme during dialysis following an aerobic and resistance intradialytic exercise programme | Intradialytic exercise programme for 4 weeks | 20 weeks | There was a significant improvement for physical function for both the standard intradialytic exercise and virtual reality exercise groups. |
| Weigmann-Faßbender et al., 2020 | Virtual Reality Exercise Programme using Nintendo® Wii outside of dialysis. | N/A | 6 weeks | No significant difference between groups was found for maximal handgrip strength or $VO^2$ peak. Daily physical activity increased, no change in HRQoL. |

(Continued)

**Table 2.** (Continued)

| Study | eHealth intervention description | Type of Comparison | Length of Follow-up | Main findings |
|---|---|---|---|---|
| Zhou et al., 2020 | Virtual reality exercise programme during dialysis. | Intradialytic exercise without virtual reality | 4 weeks | Participants reported good user experiences and it appeared as effective as standard intradialytic exercise in reducing symptoms. |
| Web-based platforms | | | | |
| Castle et al., 2022 | The web-based education platform (ExeRTiOn) contained x12 sessions including goal setting and self-management around physical activity and healthy eating. | Usual care | 52 weeks | Adherence and retention rate were both good. There were also favourable changes in body weight and physical function. |
| Donald et al., 2022 | MKMH website is a patient-facing open-access, interactive, self-management website that provides information on various lifestyle topics. | N/A | 8 weeks | Participants reported that the content of the website was acceptable. |
| Heiden et al., 2013 | The platform had three functions: Patient education regarding diet restrictions, a food analyser database, and decision support regarding phosphate binder dose. | N/A | N/A | The usability testing and qualitative interviews mainly resulted in positive feedback. |
| Humalda et al., 2020 | Self-management program dedicated to sodium restriction. Including self-regulation, self-efficacy, goal setting and social support. | N/A | 12 weeks | No significant between group difference in sodium intake. |
| Ong et al., 2022 | The development of a digital self-care Kidney Health Program that is accessed by the internet (ODYSSEE). The content of ODYSSEE Kidney Health is organized into 16 session including goal setting and behaviour change. | N/A | 16 weeks | Overall, participants reported positive satisfaction toward the navigation, layout, and content of the programme. |
| Other | | | | |
| Anand et al., 2021 | The app and the wearable (physical activity tracker) allowed participants to track their step counts. This was supplemented with telephone calls and text messages. | Activity counselling sessions and an 8-week exercise program. Weekly 1 hour exercise sessions led by fitness professionals. | 16 weeks | No difference in physical function, blood pressure or anthropometric measurements. The intervention was acceptable. |
| Naseri-Salahshour et al., 2020 | The telegram messages included educational nutritional content including descriptions of harmful food and the benefits of healthy foods for haemodialysis patients. | Usual care | 12 weeks | Significant increase in HRQoL. Decrease in potassium, serum sodium and serum phosphorus. |
| O'Brien & Meyer., 2020 | All participants were taught how to use the mobile activity tracker to monitor their daily physical activity (steps per day). This was synced with their smartphone. | N/A | 4 weeks | Participants reported only minimal problems. Many wore the activity monitor to become healthier as well as monitor their sleep and water intake. |
| Sevick et al., 2016 | Participants self-monitor their meals in a software programme on PDA to track sodium, calorie and protein intake into the BalanceLog® (Microlife, Golden, Colorado, USA) software program. | Usual care + dietary educational modules delivered during scheduled haemodialysis appointments. | 16 weeks. | No change in interdialytic body mass gain or sodium intake. Intervention was feasible and acceptable. |
| Zemp et al., 2022 | An individually tailored set of exercises. A pool of 34 single and multiple joint exercises performed twice per week. This was remotely monitored and participants were provided with a tablet. | N/A | 12 weeks. | 44% (n = 86) of participants were eligible. Out of these n = 22 agreed to participate. Adherence was 73%. No significant change in physical function. |

Blood pressure (BP), body mass index (BMI), health related quality of life (HRQoL), mean arterial pressure (MAP), not applicable (N/A), patient activation measure (PAM). *Reports of the same study so have been grouped for synthesis.

either no change or positive effects reported [26, 50, 51, 56] (Table 4). Majority of reports were of no change for body mass [26, 43, 50, 51], circulating lipids [43, 51, 54], and energy and protein intake [25, 50, 53, 56]. With positive effects reported for blood pressure [49–51] and no

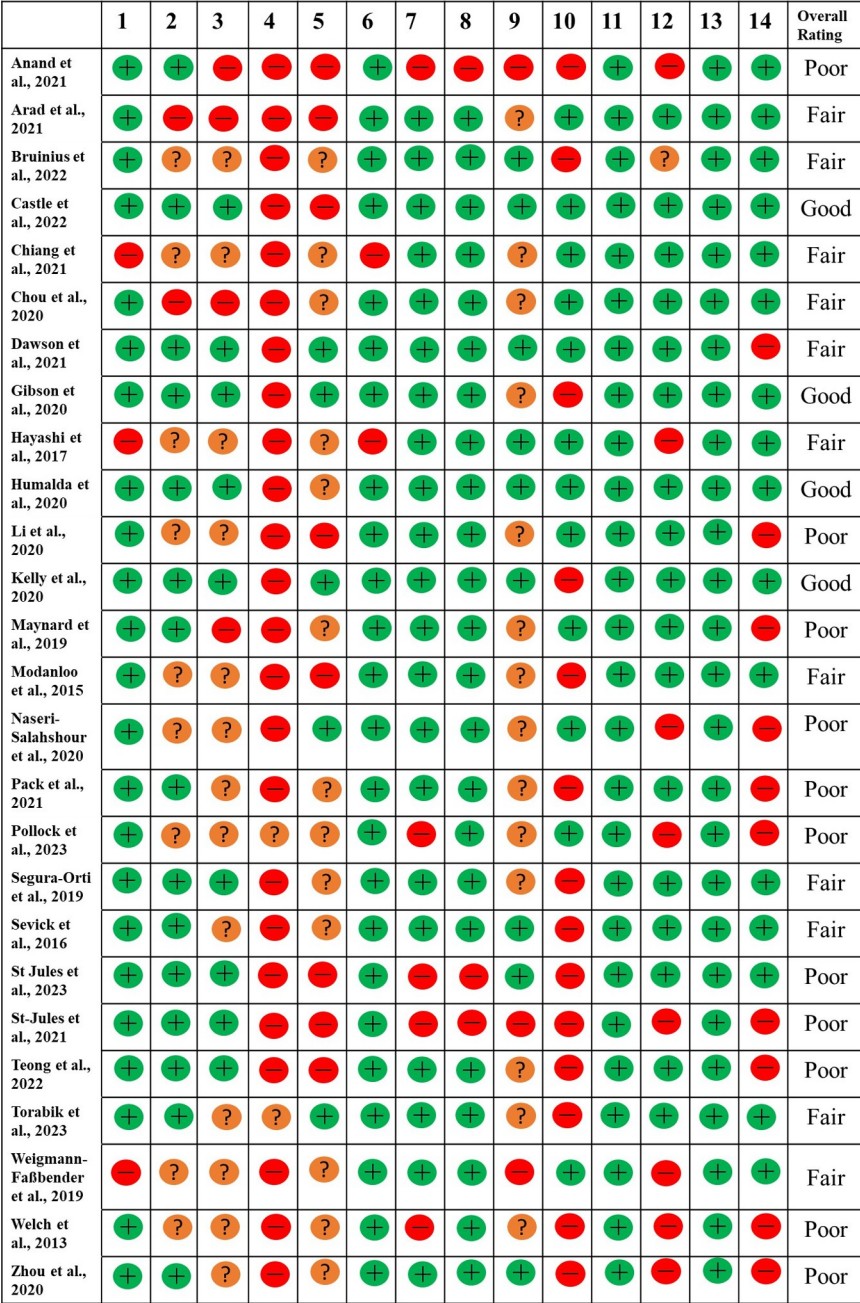

**Fig 2. Quality rating for the included randomised controlled trials & non-randomised controlled trials.** Overall quality rating was rated as poor (0–4 as "yes"), fair (5–10 as "yes"), as good (11–14 as "yes"). A "fatal flaw" defined as "no" for questions 7, 8, or 14 resulted in the study being downgraded a category (regardless of overall score).

change or positive effects reported for intradialytic weight gain [42, 48, 54, 56]. (Table 4). There were two reports for serum calcium [48, 53] and parathyroid hormone [48, 53], with reports of positive effects on serum calcium [53]. A retrospective cohort analysis showed an improvement in their composite outcome of decline in eGFR and incidence of ESKD following the use of a lifestyle application [45]. See Table 2 for the findings from the mobile application process evaluations [39, 46].

| | 1 | 2 | 3 | 4 | 5 | 6 | 7 | 8 | 9 | 10 | 11 | 12 | Overall Rating |
|---|---|---|---|---|---|---|---|---|---|---|---|---|---|
| Cueto-Manzano et al., 2015 | + | − | ? | ? | − | + | ? | ? | + | − | − | ? | Poor |
| Donald et al., 2022 | + | + | + | ? | + | + | + | ? | − | + | − | ? | Poor |
| Doyle et al., 2019 | + | + | + | ? | − | + | + | ? | + | + | − | ? | Poor |
| El Khoury et al., 2021 | + | + | + | ? | + | + | + | ? | + | + | − | ? | Fair |
| O'Brien & Meyer et al., 2020 | + | + | + | ? | + | + | + | ? | + | − | − | ? | Fair |
| Pinto et al., 2020 | + | + | + | − | ? | + | + | ? | + | + | − | ? | Fair |
| Schrauben et al., 2022 | + | + | + | ? | + | + | + | ? | − | + | − | ? | Poor |
| Zemp et al., 2022 | + | + | + | ? | + | + | + | + | − | + | − | ? | Fair |

**Fig 3. Quality rating for the uncontrolled before-after studies.** Overall rating was rated as poor (0–4 as "yes"), fair (5–8 as "yes"), as good (9–12 as "yes"). A "fatal flaw" was defined as "no" for either questions 5 or 9.

### Short-messaging service (SMS)

Three trials [27, 58, 60] reported that the SMS intervention was feasible and acceptable, with one [27] reporting a positive effect on HRQoL (Table 5). There was no effect found for diet quality [27, 60], and mixed effects for adherence to renal dietary recommendations [57, 60] (Table 5). Two [57, 60] out of three [27, 57, 60] trials reported a positive effect on serum phosphate (Table 6). Overall the effect of the intervention on blood pressure [27, 60], body mass [27, 61], serum albumin [57, 60], serum bicarbonate [27, 60] and serum potassium [27, 57, 60]

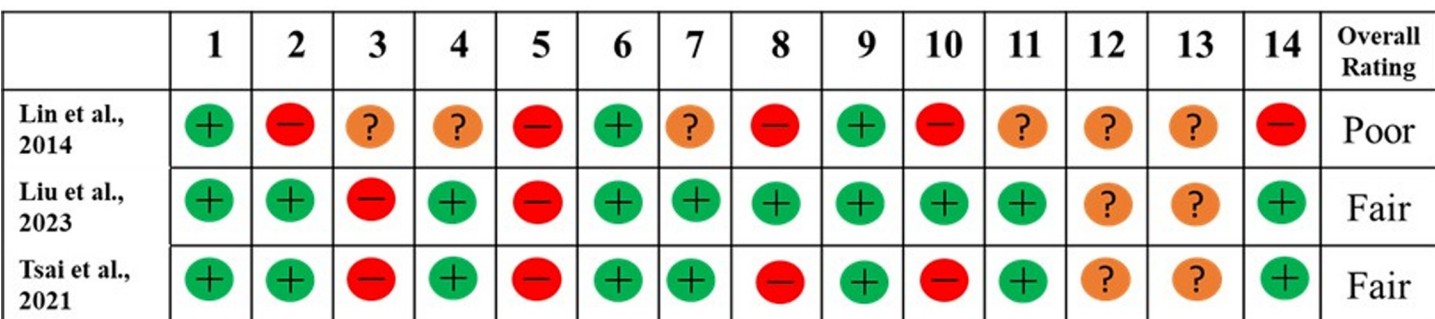

| | 1 | 2 | 3 | 4 | 5 | 6 | 7 | 8 | 9 | 10 | 11 | 12 | 13 | 14 | Overall Rating |
|---|---|---|---|---|---|---|---|---|---|---|---|---|---|---|---|
| Lin et al., 2014 | + | − | ? | ? | − | + | ? | − | + | − | ? | ? | ? | − | Poor |
| Liu et al., 2023 | + | + | − | + | − | + | + | + | + | + | + | ? | ? | + | Fair |
| Tsai et al., 2021 | + | + | − | + | − | + | + | − | + | − | + | ? | ? | + | Fair |

**Fig 4. Quality rating for the cohort and cross-sectional studies.** Overall quality rating was rated as poor (0–4 as "yes"), fair (5–10 as "yes"), as good (11–14 as "yes").

**Table 3. Visual representation of reported effect direction of dietary and lifestyle mobile applications on participant reported outcomes.**

| Study | Design | Population | Acceptability | Feasibility/Usability | HRQoL | Knowledge | Self-Efficacy | Self-Management | Quality Assessment |
|---|---|---|---|---|---|---|---|---|---|
| Li et al., 2020 | RCT | CKD 1–5 | | | ▲ | | ▲ | ▲ | Poor |
| Pack et al., 2021 | RCT | HD | | | ▲ | | ▲ | | Poor |
| St-Jules et al.,2021 | RCT | HD | ▲ | ▲ | | | | | Poor |
| Teong et al., 2022 | RCT | HD | | | | ▲ | | | Poor |
| Welch et al., 2013 | RCT | HD | ▲ | | | | ◄► | | Poor |
| Chiang et al., 2021 | NRCT | HD | | | | ▲ | ▲ | | Fair |
| El Khoury et al., 2021 | UCBA | HD | | ◄► | | ▲ | | | Fair |
| Hayashi et al., 2017 | NRCT | HD | ▲ | ▲ | ◄► | | | | Fair |
| Lin et al., 2014 | CST | CKD 1–5 | ▲ | ▲ | | | | | Poor |
| Pinto et al., 2020 | NRCT | HD | | ▲ | ◄► | | | ▲ | Fair |
| Tsai et al., 2021 | NRCT | CKD 1–5 | | | | ▲ | | | Fair |

The effect of dietary and lifestyle mobile applications on participant reported outcomes. Study design: randomised control trial (RCT); non-randomised controlled trial (NRCT); uncontrolled before-after study (UCBA); cross-sectional study (CST). Population: haemodialysis (HD); chronic kidney disease stages 1–5 (CKD 1–5). Effect direction: ▲ = positive health effect; ▼ = negative health effect; ◄► = no change/mixed effects/conflicting findings. NIH study quality rating: denoted by row colour: amber = fair; red = poor.

was unclear (Table 6). Two trials [57, 60] reported an improvement in medication including adherence [57], and decreases in phosphate binders [60], whilst another no difference in anti-hypertensives [27] (Table 6).

## Videoconferencing/online videos

Two studies employed video technologies to deliver health coaching [62] or supervised exercise [78], they were found to be feasible and acceptable [62], and they improved some outcomes such as HRQoL [62] and six-minute walking distance [78]. Process evaluations have also shown that video technologies are implementable [63, 64] (Table 1).

## Virtual reality exercise

Three RCTs [66, 67, 69] and two non-RCTs [65, 68] investigated the effect of VREx on a range of outcomes (Table 1). Two trials reported gait speed [66, 67], with one reporting no effect compared to usual care [66], and another reporting increases in both virtual reality and standard intradialytic training groups [67]. The aforementioned trial [67] also reported similar findings for the STS-10 and STS-60 with improvement following VREx, although not compared to a standard intradialytic exercise group. There were positive effects reported for physical activity [66, 68], but there appeared no effect for depressive symptoms [66, 67].

## Web-based platforms

There were two RCTs [70, 72], and one un-controlled before-after study [71] which investigated the effect of web-based platform, where participants completed sessions aimed at delivering either lifestyle [70, 71] or dietary [72] content. The dietary trial [72] involved a 3-month web-based self-management programme dedicated to sodium restriction, they found that urinary sodium excretion decreased (compared to baseline) following a 9-month follow-up [72]. The lifestyle interventions investigated the effect of web-based education platform for 8 [71] and 12-weeks [70], the trials independently reported positive effects on body mass and physical function [70], and were understandable and acceptable to participants [71]. These findings

**Table 4. Visual representation of reported effect direction of dietary and lifestyle mobile applications on clinical outcomes.**

| Study | Design | Population | Albumin | Body Mass | Blood Pressure | Circulating Lipids | Energy and Protein Intake | Intradialytic Weight Gain | Phosphorus/ Phosphate | Potassium | Sodium | Quality Assessment |
|---|---|---|---|---|---|---|---|---|---|---|---|---|
| Li et al., 2020 | RCT | CKD 1–5 | | ◄► | | ◄► | | | | | | Poor |
| Pack et al., 2021 | RCT | HD | ◄► | | | | | | ▲ | ▲ | | Poor |
| Pollock et al., 2023 | RCT | TP | | | ▲ | | | | | | | Poor |
| St-Jules et al., 2023 | RCT | CKD 1–5 | | ◄► | ◄► | ◄► | | | ◄► | | ◄► | Poor |
| St-Jules et al., 2021 | RCT | HD | ◄► | | | | | | ◄► | | | Poor |
| Teong et al., 2022 | RCT | HD | ◄► | | | | ◄► | | ◄► | | | Poor |
| Torabikhah et al., 2023 | RCT | HD | | | | ▲ | | ▲ | ▲ | ▲ | | Fair |
| Welch et al., 2013 | RCT | HD | | | | | ▼ | ▲ | ▲ | ▲ | ▲ | Poor |
| Chiang et al., 2021 | NRCT | HD | ◄► | | | | | | ◄► | | | Fair |
| El Khoury et al., 2021 | UCBA | HD | ◄► | ◄► | | | ▲ | | ◄► | ◄► | ▲ | Fair |
| Hayashi et al., 2017 | NRCT | HD | | | | | | ◄► | ◄► | ◄► | | Fair |
| Pinto et al., 2020 | UCBA | HD | | | | | | ◄► | ▲ | ◄► | | Fair |
| Schrauben et al., 2022 | UCBA | CKD 1–5 | ◄► | ▲ | ▲ | | ◄► | | ▲ | ▲ | ◄► | Poor |

The effect of dietary and lifestyle mobile applications on clinical outcomes. Study design: randomised control trial (RCT); non-randomised controlled trial (NRCT); uncontrolled before-after study (UCBA). Population: haemodialysis (HD); kidney transplant recipients (TP); chronic kidney disease stages 1–5 (CKD 1–5). Effect direction: ▲ = positive health effect; ▼ = negative health effect; ◄► = no change/mixed effects/conflicting findings. NIH study quality rating: denoted by row colour: amber = fair; red = poor.

around the acceptability of these interventions are congruent with the findings of two developmental and qualitative studies [29, 30].

## Other eHealth interventions

An RCT [73] investigated the effect of a wearable on physical activity levels, they reported that the intervention was feasible, although steps per day decreased in the intervention group, the minutes of moderate to vigorous physical activity increased. There appeared to be no effect on HRQoL and mental health measures during the study [73]. A similar study [75] found a wearable activity tracker was feasible and acceptable in encouraging physical activity in older transplant recipients. A study that employed a web-based tablet application to deliver and record a 12-week physical activity programme was found to be feasible [77]. Two RCTs aimed at assessing dietary interventions delivered by telegram messages [74], and through a PDA [76], were shown to improve HRQoL [74], and be acceptable and feasible to participants [76].

## Protocol publications

A description of the included eight protocol publications is provided in S3 Table.

**Table 5. Visual representation of reported effect direction of short messaging service (SMS) on participant reported outcomes.**

| Study | Design | Population | Acceptability | Dietary Guidelines Adherence | Diet Quality | Feasibility | HRQoL | Quality Assessment |
|-------|--------|-----------|---------------|------------------------------|--------------|-------------|-------|--------------------|
| Arad et al., 2021 | RCT | HD | | ◄► | | | | Fair |
| Bruinius et al., 2022 | RCT | CKD 1–5 | ▲ | | | ▲ | | Fair |
| Dawson et al., 2021 | RCT | HD | ▲ | ▲ | ◄► | ▲ | ▲ | Fair |
| Kelly et al., 2020 | RCT | CKD 1–5 | ▲ | | ◄► | ▲ | ◄► | Good |

The effect of dietary and lifestyle mobile applications on clinical outcomes. Study design: randomised control trial (RCT). Population: haemodialysis (HD); chronic kidney disease stages 1–5 (CKD 1–5). Effect direction: ▲ = positive health effect; ▼ = negative health effect; ◄► = no change/mixed effects/conflicting findings. NIH study quality rating: denoted by row colour: green = good; amber = fair.

## Discussion

This is the first review that has aimed to scope the available literature to understand the type of eHealth interventions that have been employed to deliver lifestyle interventions in the CKD population. The majority were mobile applications which included both dietary and physical activity components. There was also a number of interventions that used SMS messages, video-conferencing, virtual reality and web-based platforms to deliver lifestyle interventions. There was considerable heterogeneity with regards to the study outcomes reported. These eHealth interventions appeared acceptable and feasible to participants with some tentative evidence that mobile applications may have an effect on blood pressure, intradialytic weight gain, potassium and sodium, although these effects were not observed in all included reports and are therefore far from definitive. Moreover, the aim of the current review was to scope the literature rather than prove efficacy. We assessed the quality of the included RCTs, allocation concealment, blinding (of participants, providers and outcome assessment), the reporting of adherence, and the avoidance of other interventions were noted as common concerns across studies and should be addressed in the design and conduct of future studies. This resulted in many of the included studies in the current review being rated as poor quality.

Similarly to this review (and another recent review in the diabetes population [79]), the previous Cochrane review assessing the effect of eHealth interventions in the CKD population reported the majority of included studies were mobile or tablet applications alongside electronic monitoring and were acceptable to participants [13]. They reported that eHealth interventions may improve the management of dietary sodium intake and fluid management [13] which aligns with our findings. In agreement with our review, they found poor methodological quality of the included studies which limited the translation of the findings [13]. Another systematic review investigated the effect of nutritional mobile applications on a range of outcomes in the CKD population [80]. They reported that mobile applications have the potential to help

**Table 6. Visual representation of reported effect direction of short messaging service (SMS) on clinical outcomes.**

| Study | Design | Population | Albumin | Bicarbonate | Blood Pressure | Body Mass | Medication | Phosphate | Potassium | Quality Assessment |
|-------|--------|-----------|---------|-------------|----------------|-----------|------------|-----------|-----------|--------------------|
| Arad et al., 2021 | RCT | HD | ▲ | | | | ▲ | ▲ | ▲ | Poor |
| Dawson et al., 2021 | RCT | HD | ◄► | ◄► | ◄► | | ▲ | ▲ | ◄► | Fair |
| Kelly et al., 2020 | RCT | CKD 1–5 | | ◄► | ◄► | ▲ | ◄► | ◄► | ◄► | Good |
| Modanloo et al., 2015 | RCT | HD | | | | ◄► | | | | Fair |

The effect of dietary and lifestyle mobile applications on clinical outcomes. Study design: randomised control trial (RCT). Population: haemodialysis (HD); chronic kidney disease stages 1–5 (CKD 1–5). Effect direction: ▲ = positive health effect; ▼ = negative health effect; ◄► = no change/mixed effects/conflicting findings. NIH study quality rating: denoted by row colour: green = good; amber = fair; red = poor.

improve adherence to dietary restrictions pertaining to sodium, potassium, phosphorus, protein, calories and fluid within CKD [80], in like manner to our review. However, they only included two RCTs in the review [80], the present review included eight RCTs [43, 47, 49, 51–54, 56] assessing dietary or lifestyle mobile applications reporting outcomes such phosphorus, potassium and sodium, and found some improvement in these outcomes. Assessing the effect of dietary mobile applications on circulating clinical markers such as those aforementioned may be an area for an updated systematic review (and meta-analysis).

The majority of the included e-Health interventions in our review were in the ESKD (dialysis and transplant recipient) population. There should be further investigation of these interventions in early stages CKD, as the modifications of lifestyle factors is one of the cornerstones of disease management, and may positively impact disease progression and outcomes. Lifestyle interventions being delivered by e-Health may also have to be tailored for these different CKD populations, for example dietary protein recommendations vary dependent on CKD stage [4]. Physical activity and/or exercise components may also be required to be adapted based on CKD stage to ensure that there are options for all individuals to become active.

Given the association between increased physical activity levels and mortality in the CKD population [81], there were only five studies which examined this outcome [41, 62, 66, 68, 73]. The previous review from 2019 did not include any studies reporting physical activity as an outcome [13]. Given that the CKD population is largely sedentary and changing behaviour is a major barrier to individuals becoming more active, future trials should endeavour to test new digital technology and eHealth interventions aimed at increasing activity levels. The present review included a number of studies of VREx. For many of the reported outcomes VREx was effective, although not superior to standard (intradialytic) exercise programmes. Standard exercise programmes may not be the most appropriate option to engage all individuals with CKD in physical activity, and therefore there may be some individuals who would benefit from VREx, although further testing is required.

The preponderance of studies demonstrated that such technologies are feasible and acceptable to participants. Future work should aim to test the efficacy of these interventions on relevant outcomes in appropriately designed and powered RCTs. The co-development and design of these interventions should occur alongside participants and key-stakeholders [15]. Moreover, these trials should include implementation work or pragmatic and adaptive designs that take into account feasibility, as the widespread uptake of eHealth interventions is currently limited [15]. That said, there were two included protocol publications [35, 36] for appropriately powered RCTs investigating the effect and engagement of web-based self-management platforms, they were co-designed with relevant stakeholders so that they can be embedded as part of usual clinical care should they prove efficacious. There was no cost-effectiveness analysis included in the current review, although eHealth interventions may have the potential to reduce cost in the long-term they will undoubtedly require upfront investment. Resultantly cost-analysis alongside efficacy work will be crucial and may increase the chances of widespread adoption.

Many of the eHealth interventions did not include participant information regarding the socioeconomic background and health literacy levels, these individuals have the worse health outcomes and are the least likely to access and benefit from eHealth interventions. Future eHealth research should investigate ways to engage these individuals. There was heterogeneity in the intervention components of the included studies and a large number of outcomes reported in the included studies which limited our narrative synthesis. Future research should report the core (minimum) outcomes as outlined by the SONG collaboration [82]. Lack of, and poor reporting of allocation concealment, blinding and adherence was a common theme

in the included quality assessment. Some of the deficiencies in reporting could be addressed by following the CONSORT statement [83].

## Conclusion

The use of eHealth interventions will only grow and there is likely be a move towards new technologies (such as artificial intelligence and virtual reality). However, currently there is insufficient evidence to make recommendations for specific lifestyle eHealth interventions to be implemented into clinical care in the CKD populations. Properly powered RCTs which not only demonstrate efficacy, but also address barriers to implementation are needed to enhance widespread adoption.

## Supporting information

**S1 Checklist. PRISMA-ScR checklist.**
(DOCX)

**S1 File. Sample search strategy for MEDLINE.**
(DOCX)

**S1 Table. Criteria used to assess the quality of the included studies.**
(DOCX)

**S2 Table. Relevant study and trial registrations excluded from synthesis.**
(DOCX)

**S3 Table. Included protocol publications.**
(DOCX)

## Acknowledgments

We thank Selina T. Lock (University of Leicester Library) for her advice with the database searching.

## Author Contributions

**Conceptualization:** Daniel S. March.

**Data curation:** Daniel S. March.

**Formal analysis:** Daniel S. March.

**Investigation:** Ayesha Butt, Harsimran K. Dhaliwal, Matthew M.P. Graham-Brown, Rishika Rawat, Thomas J. Wilkinson, Daniel S. March.

**Methodology:** Ffion Curtis, Harsimran K. Dhaliwal, Rishika Rawat, Thomas J. Wilkinson.

**Project administration:** Daniel S. March.

**Resources:** Ffion Curtis, Ayesha Butt, Harsimran K. Dhaliwal, Rishika Rawat, Thomas J. Wilkinson, Daniel S. March.

**Supervision:** Ffion Curtis, James O. Burton.

**Validation:** Ffion Curtis, Daniel S. March.

**Writing – original draft:** Daniel S. March.

**Writing – review & editing:** Ffion Curtis, Harsimran K. Dhaliwal,
   Matthew M.P. Graham-Brown, Courtney J. Lightfoot, Alice C. Smith,
   Thomas J. Wilkinson, Daniel S. March.

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
