## [Decision Letter · Decision Letter 0]

28 Nov 2023

PONE-D-23-29466Lifestyle interventions delivered by eHealth in Chronic Kidney Disease: A Scoping ReviewPLOS ONE

Dear Dr. March,

Thank you for submitting your manuscript to PLOS ONE. After careful consideration, we feel that it has merit but does not fully meet PLOS ONE’s publication criteria as it currently stands. Therefore, we invite you to submit a revised version of the manuscript that addresses the points raised during the review process.

We look forward to receiving your revised manuscript.

Kind regards,

Henry H.L. Wu, MBChB PGCert

Academic Editor

PLOS ONE

Journal Requirements:

"No authors have competing interests."

4. We note that you have referenced (unpublished) on page 5, which has currently not yet been accepted for publication. Please remove this from your References and amend this to state in the body of your manuscript: (ie “Bewick et al. [Unpublished]”) as detailed online in our guide for authors

Additional Editor Comments:

Thank you for your submission.

As both reviewers have pointed out, this is a very comprehensively conducted scoping review summarizing the topic, and allows for the highlighting of important knowledge gaps and future research directions in the field of EHealth within CKD and Nephrology.

Please refer to and consider the reviewer comments in making the revision

Reviewers' comments:

Reviewer's Responses to Questions

**Comments to the Author**

1. Is the manuscript technically sound, and do the data support the conclusions?

Reviewer #1: Yes

Reviewer #2: Yes

2. Has the statistical analysis been performed appropriately and rigorously? 

Reviewer #1: Yes

Reviewer #2: Yes

3. Have the authors made all data underlying the findings in their manuscript fully available?

Reviewer #1: Yes

Reviewer #2: Yes

4. Is the manuscript presented in an intelligible fashion and written in standard English?

Reviewer #1: Yes

Reviewer #2: Yes

5. Review Comments to the Author

Reviewer #1: Thank you for this well written scoping review I have a few comments.

In the introduction I would think some context of the target population would be worthy, demographics of CKD including relationship with age, co-morbidities and ethnicity. This is important as interventions being reviewed need to be applicable to different populations. You note this in the discussion briefly mentioning health literacy but maybe some reference in the intro as well.

Results

Page 7 line 170 You say 54 studies included with 22 full text studies ? Its not very clear I think you mean 23 RCT's- see abstract it says 23 of which only 22 had full text.

General comment

The results are presented as per intervention, but you included studies of CKD 1-5 and ESKD and the population within that range will have different needs in terms of interventions. It seems the majority of studies were in the haemodialysis population and less in early CKD, no mention of peritoneal dialysis and little transplantation. I think some reference to this in the discussion would be worthy as early CKD is not well represented but may have the greatest gain? Would it be worth mentioning how different interventions would need to be specific to the different stages of CKD; for example sodium and fluid balance maybe more appropriate for later stages. Also highlight some of the difficulties with exercise and how this needs to be tailored to stages of CKD, age etc

Page 24 line 277-278 just some grammar change needed.

Discussion- is there any reviews in the non CKD population worthy of a mention such as diabetes where these types of interventions will have been studied and the commonality of multiple co-morbidities in the CKD population which maybe transferable.

Minor comment in conclusion maybe bullet point recommendations for future research.

Reviewer #2: There is indeed a gap in knowledge with respect to Ehealth and health promotion esp in CKD, the area addressed in this scoping review provided a good summary of what is currently available.

The analytical method is robust and meticulously thought through with appropriate conclusion.

I believe this review is a good precursor for systematic review in time to come when digital health is more established.

Congratulation on a job well done!

6. PLOS authors have the option to publish the peer review history of their article (what does this mean?). If published, this will include your full peer review and any attached files.

Reviewer #1: No

Reviewer #2: No

---

## [Author Response · Author response to Decision Letter 0]

15 Dec 2023

Reviewer 1 

In the introduction I would think some context of the target population would be worthy, demographics of CKD including relationship with age, co-morbidities and ethnicity. This is important as interventions being reviewed need to be applicable to different populations. You note this in the discussion briefly mentioning health literacy but maybe some reference in the intro as well. 

Thank you for this suggestion. We agree that this is an important point to add to the introduction. Please see Lines 55-57. We have added ‘Furthermore, CKD is more common in older age, and is associated with multiple long-term conditions (10), and low levels of health literacy (11), which further complicates the implementation process.’

Reviewer 1 

Results - Page 7 line 170 You say 54 studies included with 22 full text studies ? Its not very clear I think you mean 23 RCT's- see abstract it says 23 of which only 22 had full text.

Apologies for the confusion. We have clarified this. Please see line 174. We hope this reads clearer. If further changes are required then please do let us know.

Reviewer 1 

The results are presented as per intervention, but you included studies of CKD 1-5 and ESKD and the population within that range will have different needs in terms of interventions. It seems the majority of studies were in the haemodialysis population and less in early CKD, no mention of peritoneal dialysis and little transplantation. I think some reference to this in the discussion would be worthy as early CKD is not well represented but may have the greatest gain? Would it be worth mentioning how different interventions would need to be specific to the different stages of CKD; for example sodium and fluid balance maybe more appropriate for later stages. Also highlight some of the difficulties with exercise and how this needs to be tailored to stages of CKD, age etc

Thank you for this very important point. Please see lines 352-356 where we have addressed the lower number of studies in early-stage CKD. Furthermore, we have discussed how lifestyle interventions delivered by e-Health may need to be tailored depending on CKD stage (lines 356-360).

Reviewer 1 

Page 24 line 277-278 just some grammar change needed.

Thank you for this. Please see Line 282 where we have amended this.

Reviewer 1

Discussion- is there any reviews in the non CKD population worthy of a mention such as diabetes where these types of interventions will have been studied and the commonality of multiple co-morbidities in the CKD population which maybe transferable. 

Please see Line 335 where we have added a very recent reference for a review in the diabetes population which reported similar findings to ours.

Reviewer 1

Minor comment in conclusion maybe bullet point recommendations for future research.

Thank you for this comment. We have addressed future research within Lines 397-402 of the conclusion. Although we don’t feel that bullet points would work particularly well here. Although we would be happy to amend if the Editor felt appropriate.

Reviewer 2 

There is indeed a gap in knowledge with respect to Ehealth and health promotion esp in CKD, the area addressed in this scoping review provided a good summary of what is currently available. The analytical method is robust and meticulously thought through with appropriate conclusion. I believe this review is a good precursor for systematic review in time to come when digital health is more established. Congratulation on a job well done!

We thank Reviewer 2 for their kind words regarding this piece of work. We agree that this Scoping Review sets the scene for a future full Systematic Review in this area. The research is this area is growing exponentially, and the appropriate time for a full Systematic Review will undoubtedly be soon.

---

## [Decision Letter · Decision Letter 1]

27 Dec 2023

Lifestyle interventions delivered by eHealth in Chronic Kidney Disease: A Scoping Review

PONE-D-23-29466R1

Dear Dr. March,

We’re pleased to inform you that your manuscript has been judged scientifically suitable for publication and will be formally accepted for publication once it meets all outstanding technical requirements.

Kind regards,

Henry H.L. Wu, MBChB PGCert

Academic Editor

PLOS ONE

Additional Editor Comments (optional):

The authors have now satisfied the queries from the first review.

Reviewers' comments:

Reviewer's Responses to Questions

**Comments to the Author**

1. If the authors have adequately addressed your comments raised in a previous round of review and you feel that this manuscript is now acceptable for publication, you may indicate that here to bypass the “Comments to the Author” section, enter your conflict of interest statement in the “Confidential to Editor” section, and submit your "Accept" recommendation.

Reviewer #1: All comments have been addressed

2. Is the manuscript technically sound, and do the data support the conclusions?

Reviewer #1: Yes

3. Has the statistical analysis been performed appropriately and rigorously? 

Reviewer #1: Yes

4. Have the authors made all data underlying the findings in their manuscript fully available?

Reviewer #1: Yes

5. Is the manuscript presented in an intelligible fashion and written in standard English?

Reviewer #1: Yes

6. Review Comments to the Author

Reviewer #1: Thank you for taking on board the comments and making the changes to the article I have no further comments

7. PLOS authors have the option to publish the peer review history of their article (what does this mean?). If published, this will include your full peer review and any attached files.

Reviewer #1: No

---

## [Editor Report · Acceptance letter]

11 Jan 2024

PONE-D-23-29466R1 

PLOS ONE

Dear Dr. March, 

I'm pleased to inform you that your manuscript has been deemed suitable for publication in PLOS ONE. Congratulations! Your manuscript is now being handed over to our production team.

Kind regards, 

on behalf of

Dr. Henry H.L. Wu 

Academic Editor

PLOS ONE